# Out-of-Hospital Cardiac Arrest during the COVID-19 Pandemic: A Systematic Review

**DOI:** 10.3390/healthcare11020189

**Published:** 2023-01-08

**Authors:** Amreen Aijaz Husain, Uddipak Rai, Amlan Kanti Sarkar, V. Chandrasekhar, Mohammad Farukh Hashmi

**Affiliations:** 1School of Pharmaceutical and Population Health Informatics, DIT University, Dehradun 248009, India; 2Institution of Clinical Research India (ICRI), Mumbai 400093, India; 3Mahatma Gandhi Memorial Hospital, Warangal 506002, India; 4Department of Electronics and Communication Engineering, National Institute of Technology, Warangal 506004, India

**Keywords:** out-of-hospital cardiac arrest (OHCA), cardiopulmonary resuscitation (CPR), emergency medical services (EMS), coronavirus disease 2019 (COVID-19)

## Abstract

*Objective*: Out-of-hospital cardiac arrest (OHCA) is a prominent cause of death worldwide. As indicated by the high proportion of COVID-19 suspicion or diagnosis among patients who had OHCA, this issue could have resulted in multiple fatalities from coronavirus disease 2019 (COVID-19) occurring at home and being counted as OHCA. *Methods*: We used the MeSH term “heart arrest” as well as non-MeSH terms “out-of-hospital cardiac arrest, sudden cardiac death, OHCA, cardiac arrest, coronavirus pandemic, COVID-19, and severe acute respiratory syndrome coronavirus 2 (SARS-CoV-2).” We conducted a literature search using these search keywords in the Science Direct and PubMed databases and Google Scholar until 25 April 2022. *Results*: A systematic review of observational studies revealed OHCA and mortality rates increased considerably during the COVID-19 pandemic compared to the same period of the previous year. A temporary two-fold rise in OHCA incidence was detected along with a drop in survival. During the pandemic, the community’s response to OHCA changed, with fewer bystander cardiopulmonary resuscitations (CPRs), longer emergency medical service (EMS) response times, and worse OHCA survival rates. *Conclusions*: This study’s limitations include a lack of a centralised data-gathering method and OHCA registry system. If the chain of survival is maintained and effective emergency ambulance services with a qualified emergency medical team are given, the outcome for OHCA survivors can be improved even more.

## 1. Introduction

The death toll resulting from the emerging COVID-19 pandemic is expected to be much higher due to the SARS-CoV-2 virus’s direct effect [1,2]. COVID-19 has been linked to a number of time-critical events, including “out-of-hospital cardiac arrest (OHCA)” [3]. In comparison to previous years, a recent comprehensive study found almost a two-fold jump in OHCA rates, prolonged duration for ambulance response, and a 33% drop in the probability of survival of OHCA cases related to the pandemic season [4,5]. 

COVID-19’s indirect effects on OHCA could have been caused by a variety of reasons. Patients seem to be less inclined to report to a nearby hospital for emergency medical care, particularly in cases of cardiac difficulties, and this has been suggested as a possible cause of higher occurrence [6,7]. This may raise the chances of OHCA [1,4,8]. Ambulance access and operations may potentially alter significantly, including the impact on “ambulance response time” due to changes in emergency medical services (EMS)” caseload and a delayed response in outpatient therapy due to restrictions for paramedics to wear COVID-19 “personal protective equipment (PPE)” [1,9,10]. Overall EMS response to OHCA differed by less than a minute between pre-COVID-19 and COVID-19 time periods [11,12]. In the 2020 group, the average EMS response time to OHCA was 56 s longer. In the 2020 group, the time from the call to emergency medical technician (EMT) departure rose by 39.8 s. Between the two periods, overall transfer times remained similar. EMTs in the 2020 group, on the other hand, stayed longer at the site, implying that the time required for resuscitation in the field was observed to be prolonged in the 2020 group. Figure 1 presents the comprehensive time difference of the prehospital phase between the 2019 and 2020 groups according to Jiun Hao’s research [13].

OHCA is indeed a life-threatening medical issue. The clinical outcomes of OHCA rely heavily on a very good “chain of survival” [14,15]. This aspect of the “chain of survival” entails “bystander cardiopulmonary resuscitation” (BCPR), the utilization of “automated external defibrillators” (AEDs), and “EMS”. The unusual coronavirus pandemic, on the other hand, had an unknown influence on EMS resources. It was thought to have interrupted the “chain of survival” in the prehospital emergency setting, particularly regarding layman or bystander action [2]. From the perspective of OHCA survival, institutional limits on movement may make it less likely for OHCA to manifest in a public setting. As a result, bystander “cardiopulmonary resuscitation” (CPR) and “community defibrillation” are less likely to be performed before an ambulance arrives [16].

Because community and emergency services processes have a greater impact on survival than “advanced hospital-based interventions”, there is a lot of scientific and population health concern about how they were changed. Only those who received prompt prehospital treatments benefited from sophisticated hospital-based interventions [17,18]. Evaluating the COVID-19 pandemic’s influence on OHCA treatment plans is critical for future public healthcare programmes and policies aimed at improving OHCA outcomes after the COVID era is over and for future pandemic emergency advance preparedness.

Research examining the impact of COVID-19 on the “chain of survival,” especially that of early bystander action, has produced mixed results [19,20]. As per studies by Marijon et al., bystanders were even more cautious and unwilling to perform CPR in prospective COVID-19 instances. This was presumably because bystanders were avoiding performing CPR on probable COVID-19 suspected cases [7].

This was especially worrying because BCPR combined with early defibrillation has been shown to increase a victim’s probability of survival [21]. However, some additional studies did not find these results, thus requiring a review of the already published literature on BCPR before as well as during COVID-19. There is also a need to research relevant variables, such as OHCA at home, bystander AED usage, and observed OHCA [22,23]. Prehospital “chain of survival” and EMS treatment plans are also important variables in improving OHCA outcomes. EMS requirements, EMS resuscitation procedures and durations, or the administration of supraglottic airway devices, endotracheal intubation, epinephrine, amiodarone, and mechanical CPR are all examples of advanced life support methods [23].

These characteristics are crucial in the management and care of OHCA patients. It is an early step before they are admitted to the hospital for specialized care [24]. Lim et al. earlier hypothesised that “during the outbreak of COVID-19, EMS call to arrival times were likely to increase given the challenges like heightened personal protective equipment (PPE) requirements” [4]. The influence of COVID-19 on EMS quality care, on the other hand, remains unknown [25].

The purpose of this study is to look at the epidemiology, the impact of COVID-19 outbreaks on OHCA patient care, and potential factors that influenced OHCA during the pandemic. While there have been earlier evaluations on similar themes, none of the studies has looked at the influence of various therapies on the epidemiology and frequency of OHCA during the pandemic. In this systematic review, we wanted to provide a concise overview of potential changes in OHCA profiles in a variety of pandemic-related domains as well as an assessment of the logistical challenges and concerns faced by first responders dealing with prehospital sudden cardiac events.

## 2. Search Methodology

The Preferred Reporting Items for Systematic Reviews and Meta-Analyses (PRISMA) standards were followed in compiling this systematic review. Medical Subject Headings thesaurus (MeSH) terms and keywords from relevant literature were used to build a search strategy that covers all relevant papers. We used the MeSH term “heart arrest” as well as the non-MeSH terms “out-of-hospital cardiac arrest, sudden cardiac death, OHCA, cardiac arrest, coronavirus pandemic, COVID-19, and SARS-CoV-2”. We conducted a literature search using these search keywords in the Science Direct and PubMed databases and Google Scholar until 25 April 2022. Searching whole texts retrieved much irrelevant research so the “abstracts only” approach was used. This type of approach resulted in the acquisition of the most relevant research material. The authors independently assessed the titles and abstracts for inclusion. We also looked through the reference tracking of bibliographies and manual searches during the first search to see if there were any additional studies that were relevant. Figure 2 shows search strategy for systematic review of OHCA during COVID-19 pandemic.

All the relevant search results were downloaded and exported in CSV (comma-separated values) format. This data were imported into an Excel file named “search data”. All the titles and relevant information were sorted in separate sheets with the name of the databases and the number of relevant results found through each database. The data were combined in another Excel sheet named “combined”, and duplicate research data were highlighted. These duplicates were then removed using Excel’s automated function.

The following inclusion criteria were applied to all studies: (A) patients who developed OHCA during COVID-19 pandemic and (B) publications that reported any of the following outcomes and clinical features: incidence of OHCA during the pandemic, epidemiology of OHCA during COVID-19, mortality, return of spontaneous circulation (ROSC), filed termination of resuscitation (TOR), survival to hospitalization, survival to hospital discharge, aetiology of OHCA, and shockable rhythm. The following were the criteria for exclusion: (A) papers not published in English, (B) articles without historical control (comparing outcomes before and after the pandemic), (C) case reports, and (D) letters to editors. We did not seek to obtain non-peer-reviewed information because unpublished studies may have lower research quality than those that have been published.

Name of first author, country of sample group, year of publication, study design, study population, schedule cut-offs for time periods (i) pre-COVID-19 and (ii) during COVID-19, and sample sizes for timeframes (i) pre-COVID-19 and (ii) COVID-19 were among the study characteristics retrieved. Furthermore, we extracted patient data such as age, sex, percentage of patients with shockable rhythm, and percentage of patients with various OHCA etiologies. OHCA etiologies data, for example, included data such as medical, drowning, traumatic, overdose, and asphyxia. Finally, the data on the outcomes were extracted. Incidence, mortality, field TOR, ROSC, survival to hospital admission, and survival to hospital release were among the variables studied.

If data were ambiguous, we approached the publication’s author via email for confirmation. The relevant papers were manually checked to ensure that they did not use duplicate datasets. We found no proof of these after a thorough investigation.

## 3. Review of Statistical Analysis Used for OHCA

The meta 4.18–0 as well as meta for 2.4–0 packages in “R 3.6.3 (R Foundation for Statistical Computing, Vienna, Austria)” were used to conduct statistical analyses [26]. For the outcomes of yearly OHCA incidence and community first responder (CFR), proportional meta-analyses were performed. Meta-analyses were conducted for the following features and outcomes: mortality, ROSC, field TOR, survival to hospital admission, and survival-to-hospital discharge.

The “Freeman-Tukey double arcsine approach” was used to transform the raw data for the outcomes of yearly OHCA incidence and CFR in the meta-analyses into percentages [27]. Following that, the converted data were aggregated using an inverse variance process before being backtransformed into normalised proportions. Due to the significant heterogeneity of the study, the “DerSimonian-Laird estimator” was utilised as a “between-study variance estimator,” and a “random-effects model” was also used to predict overall pooled study values [28]. The data were then plotted in a forest with pertinent outcomes presented as proportions with 95% confidence intervals (CI). Two-proportion z-tests were used to compare the outcomes of “annual OHCA incidence” and CFR for pre-COVID-19 and COVID-19 eras. These findings were represented using box plots. Table 1 presents a summary of the literature review on the statistical analysis used for OHCA.

Yearly OHCA frequency was calculated using data only from publications that used a community data source, allowing for the interpretation of population characteristics. OHCA incidence during COVID-19 was manually calculated in cases where studies did not explicitly cite an incidence. This was true for studies that supplied case numbers and the related population-at-risk numbers. This was based on the assumption that the community at risk remained stable during the course of the study and that each individual could only provide one OHCA incident. To normalise them to yearly incidence per 100,000 population, the reference timeframe of each research was employed. Individual community frequency estimates were weighted and averaged depending on the size of the population at risk. Berdowski et al. used this method earlier in a systematic study of OHCA incidence [29].

To assess the pooled effects of COVID-19, “fixed- and random-effects models” were generally used in combination with the “Sidik-Jonkman estimator” and “Mantel–Haenszel” approach for other outcomes. It was predicated on the presence of significant interstudy heterogeneity [30]. For “mortality, field TOR, ROSC, survival to hospital admission, survival to hospital discharge, shockable rhythm, and etiologies of OHCA”, forest plots presented pooled odds ratios (OR) and a 95% CI. The p-value for two-tailed statistical significance was chosen as 0.05. To evaluate statistical heterogeneity, the I^2^ statistic was utilised. We looked for outliers where there was a lot of statistical heterogeneity (I^2^ > 50%) by running a series of “case deletion examinations” to find influential studies and then running leave-one-out assessments. Visual inspection of funnel plots and “Egger’s regression” were used to assess publication bias [31].

**Table 1 healthcare-11-00189-t001:** Summary table for literature review on statistical analysis used for OHCA.

Sr. No.	Main Author	Country	Article Type	Publication Date	Main Findings
1.	AS Jasne [32]	United States	Systematic search	2020	During the COVID-19 pandemic, hospitalizations for stroke-like symptoms dropped, but there were no alterations in stroke severity or early outcomes.
2.	CP KOVACH [33]	United States	Retrospective cohort analysis	2021	The COVID-19 pandemic has impacted the cardiac arrest survival chain in psychosocial as well as ethical ways.
3.	PH LAI [19]	United States	Observational studies	2020	Out-of-hospital cardiac arrests and mortality rates increased considerably during the COVID-19 pandemic compared to the same period of the previous year.
4.	E MARIJON [7]	France	Population-based, observational study	2020	When comparing the defined time period of the pandemic to the corresponding time period in previous years without a pandemic, a temporary two-fold rise in OHCA incidence was detected along with a drop in survival.
5.	RT FOTHERGILL [23]	United Kingdom	Observational study	2021	The incidence of OHCA increased dramatically during the first wave of the COVID-19 pandemic in London followed by a considerable drop in survival.
6.	ZJ LIM [4]	Australia	Systematic search	2020	During the COVID-19 pandemic, there were major differences in resuscitation procedures.
7.	M REDLENER [34]	United States	Observational study	2020	During the early stages of the pandemic, the rate of OOHCA increased at an exponential rate, and the EMS system in NYC saw an unprecedented need for critical care and resuscitation.
8.	B GRUNAU [35]	Canada	Survey research	2020	During the pandemic, people’s willingness to perform bystander resuscitation reduced; however, this has mitigated if simple PPE was accessible.
9.	UY-EVANADO [36]	United States	Population-based, observational study	2021	From March to May 2020, the community’s response to OHCA changed, with fewer bystander CPRs, longer EMS response times, and worse OHCA survival rates.
10.	NK GLOBER [37]	United States	Retrospective cohort analysis	2021	When compared to the previous year, total OHCA increased during the COVID-19 pandemic. Despite the fact that patient profiles remained comparable, the proportion of patients who died in hospitals reduced during the epidemic.

## 4. Impact of Healthcare System Transformation

Early in the COVID-19 pandemic, system difficulties may have resulted in higher OHCA incidence and mortality [38]. Many nonurgent cardiovascular diagnostic tests and planned operations were delayed, rescheduled, or cancelled as a result of the rapid and unprecedented reorganisation of healthcare systems to restrict face-to-face interaction and enable telemedicine [39,40,41,42,43]. These activities may have unknowingly reduced or delayed care for people who were at increased risk of encountering OHCA. Fear of getting the COVID-19 virus may have caused patients to avoid seeking treatment [44,45]. Indeed, new advanced research has shown that more than 25% of OHCA victims had contact with the healthcare system in the 90 days before the pandemic, and hospitalisation rates for acute MI, cardiac arrest, and stroke fell significantly during COVID-19 [46,47,48,49,50].

This decline in hospitalizations was accompanied by a reduction in “emergency medical services (EMS)” calls and a “doubling in EMS-attended fatalities” [32,33]. Under a nationwide “shelter-in-place” order, single-centre research in Denver found a drop in ambulance call to action and also an almost 2.2-fold greater incidence of OHCA, which the authors associated with increased myocardial infarction (MI)-related OHCA and restricted healthcare coverage [51,52]. COVID-19 has also adversely harmed low-income populations, ethnic minorities, and jailed people who already have limited access to medical care, low rates of CPR, and late EMS service [53,54,55]. Even after accounting for comorbid conditions that influence minority populations inequitably, Lai et al. found that Hispanic, Black, and Asian communities were at greater risk of pandemic-related OHCA and death. This highlights the widespread inequalities and contrasts in the US medical system both before and during COVID-19 [19].

## 5. OHCA Epidemiology during the Coronavirus Epidemic

During earlier research, OHCA was shown to be more common in local populations that were severely affected and damaged during the start of COVID-19. According to a Redlener analysis, EMS in New York City (NYC) reported a 220 percent spike in cardiac arrest emergency responses from February to April 2020 (8837) relative to February to April 2019 (4022) during the COVID-19 epidemic, peaking on 6 April with 330 reported cases in a single day, as shown in Figure 3 [34].

Researchers have also reported decreased resuscitation performance rates and higher mortality [7,19,20,56]. “ROSC” after OHCA decreased from “31 to 18 percent in Northern Italy, 25 to 11 percent in New York, and 23 to 13 percent in Paris” from last year [7,20]. During the pandemic, a global meta-analysis study of 10 research papers and 35,379 incidents of OHCA indicated a 2.2-fold rise and also a significantly lower survival rate compared to 2019 [4]. Lim and colleagues also found that OHCA occurs more frequently at home, intubation rates are lower, the prevalence of nonshockable rhythms is higher, and the survival rate of hospital admission and discharge is lower. “Increased OHCA incidence (88.5–64.1 vs. 69.7–49.8 per million residents; P 0.01), decreased rates of ROSC (23.0% vs. 29.8%; adjusted rate ratio [ARR], 0.82 [95% CI, 0.78–0.87]; P 0.01), and decreased survival to hospital discharge (6.6% vs. 6.6%)” were found in the “Cardiac Arrest Registry to Enhance Survival (CARES)” [33].

## 6. Perceptions towards Resuscitation during the Pandemic

OHCA sufferers have the best chance of survival if they receive early CPR and defibrillation. According to case reports of COVID-19 transmission during chest compressions, CPR is classified as an “aerosol-generating operation” by the World Health Organization (WHO) [57]. However, there is little evidence that CPR produces aerosols or is linked to the transmission of COVID-19 [58,59]. To limit the risk of exposure, healthcare professionals performing “aerosol-generating operations” during resuscitation of individuals with uncertain COVID-19 cases should wear personal protective equipment (PPE) [60,61]. Despite this, a significantly increased incidence of OHCA occurs at home where bystanders are unlikely to have PPE [62]. The rescuer is more often a personal contact of the patient in these situations. As a result, CPR is less likely to pose a major impact on their own risk of infection [61].

Despite the established benefits associated with CPR performed during cardiac arrest, the pandemic has instilled a fear culture. This fear can affect bystanders’ motivation to undertake resuscitation procedures due to a perceived increased danger of personal damage [63,64]. Bystanders who did not complete CPR for fear of catching COVID-19 have been mentioned in newspapers [64]. During the pandemic, a social media survey was performed with over 1300 layperson respondents from 26 countries. This survey found a decreased inclination to perform chest compressions, examine an unresponsive stranger, or use an AED [35].

However, evaluating whether risk perceptions influence bystander CPR on a comparatively larger scale has been difficult. During the pandemic, researchers in Paris and northern Italy saw a significant drop in bystander CPR, as shown in Figure 4 [36]. There has been no notable change in bystander resuscitation in New York, Seattle, or Pittsburgh [65,66]. The CARES analysis showed no significant difference in rates of observed cardiac arrest “(41.1 percent vs 43.7 percent; SD 5.4 percent), bystander CPR (47.7% vs. 46.8%; SD 1.7 percent), or bystander defibrillation (5.7 percent vs 8.1 percent; SD 9.4 percent) in the United States during the pandemic compared with the same period in 2019” [67].

According to existing data, the preferred approach to witnessed OHCA should continue to be the rapid detection of pulse and chest compressions by bystanders, using instantly available PPE or improvised PPE [33]. The readiness of the general populace to administer resuscitation is clearly shaky. To maintain a vigorous community response to OHCA, public health organisations must provide strong direction and support through CPR teaching and innovative technologies. By doing so, a potentially devastating and disheartening reduction in bystander CPR will be avoided [68,69]. Table 2 Shows a summary table for the Site and rate of OHCA and bystander CPR in 2020 as compared with 2019 [37].

## 7. Challenges for OHCA during COVID-19

During the pandemic, there has been no widely acknowledged international guidance or methods for resuscitation of OHCA patients since the level of risk of being exposed by first rescuers is still being debated. CPR is classified as “an aerosol-generating technique (AGP)” by the “World Health Organization (WHO)” [70,71]. With this in mind, most published guidelines support EMS personnel’s safety by requiring PPE or delaying resuscitation until PPE could be obtained and worn [72,73,74]. At the start of the pandemic, information from experiences with hospital resuscitation bolstered these suggestions. Early on in the outbreak, the “Centers for Disease Control and Prevention (CDC)” issued a report related to a group of health professionals in California who were exposed to a hospitalised COVID-19 patient without sufficient PPE. Out of 121 exposed healthcare workers, 43 developed symptoms, with three of the samples positive for COVID-19 [75,76]. To safeguard frontline employees performing AGP, the research recommends the use of higher-level respirators or N95 masks [77,78].

Many “National Resuscitation Councils (NRC)”, such as those in the United Kingdom, New Zealand, and the US, have advised using CPR with caution [79]. According to studies “Rescue breathing with a bag-mask device with a filter and a tight sea” was one of the precautions [60]. Considering that several first responders may not even have accessibility to all necessary COVID-19 PPE, the “Australian College of Emergency Medicine (ACEM)” strongly advised that first responders should at least wear gloves, protective goggles, and a mask before beginning the procedure of chest compressions [79]. The “American Heart Association” (AHA), the “English National Health Service (NHS)”, and the “Belgian Resuscitation Council” all issued guidelines that encouraged first responder protection by delaying resuscitation attempts until the doctor/nurse had put on PPE [80,81,82]. For infants and children, the “International Liaison Committee on Resuscitation (ILCoR)” provided more substantial recommendations, recommending that lay rescuers explore chest compressions as well as the beginning of rescue breaths and also public access defibrillation [72].

They recommend that healthcare professionals think about defibrillation before putting on PPE if the supposed benefit outweighs the potential dangers [61]. Because chest compressions procedure and the process of defibrillation were not generally considered to be aerosol-generating procedures (AGPs), the “Department of Health and Social Care (DHSC)” in England stated that EMS personnel have to perform CPR while waiting for any further assistance without the need for proper PPE [80,82]. According to DHSC guidelines, delayed resuscitation resulted in needless deaths. Couper et al. acknowledged this in a comprehensive study, indicating that delaying CPR by a few minutes decreased the chance of survival of COVID-19 patients. They also admitted that performing CPR on a COVID-19-positive or likely patient without protective equipment put healthcare professionals in danger [58].

Systematic studies have also found that none of the published studies answered the question of whether the process of defibrillation or chest compressions produces aerosols. According to Sayre et al., one medical professional per 10,000 may die as a direct result of viral transmission while performing CPR on a COVID-19 patient. However, on the other side, CPR saves nearly 300 lives per 10,000 with OHCA [65]. They said bystanders performing CPR must be postponed for correct PPE-wearing in areas with a high COVID-19 prevalence. Even when “chest compressions” were reported to be linked to viral transmission, exposure to additional airway techniques during the process of resuscitation made it comparatively difficult to pinpoint “chest compressions” as the sole source of viral transmission. Table 3 shows a summary table for some of OHCA characteristics prepandemic versus the COVID-19 pandemic [36].

## 8. Strategies for OHCA Management during COVID-19

### 8.1. Need to Reduce Risk Rate for Provider

Due to frequent interaction with symptomatic people, frontline healthcare personnel are at a high risk of developing respiratory diseases [58]. Coronavirus transmission can be reduced with proper PPE, such as N95 masks or “positive air pressure respirators”, mainly throughout “aerosol generating procedures (AGPs)” [83,84]. Individual (e.g., age) and system characteristics might influence provider risk. If their workforce has not been fully vaccinated, healthcare organisations might have to consider redoubling their ability to preserve a decent supply of PPE for AGPs. This is crucial because only complete immunisation of healthcare personnel provides a very low infection rate. CPR comprises AGPs, and healthcare provider vaccination percentages stay below 100%.

Even if healthcare providers obtain resistance to SARS-CoV-2 by vaccination, it is essential for them to continue to take precautionary measures against COVID-19 and its variations. On the other hand, the danger of the patient contracting COVID-19 and developing a serious illness is exceedingly high when compared to the considerably smaller chance of the resuscitation provider contracting COVID-19 and developing a medical condition. This risk is notably low in vaccinated as well as unvaccinated healthcare providers who work with patients while wearing AGP-specific PPE. Although current vaccinations have been shown to be effective against the wild-type SARS-CoV-2 and variations of concern, breakout infections, which are usually not fatal, may still occur. It is possible that boosters addressing developing variations of concern may be needed [85].

### 8.2. Offer Timely Care and Limit Provider Exposure

Chest compressions must not be deferred in individuals with probable or confirmed COVID-19 who have suddenly stopped breathing [86]. Chest compressions can be conducted using a chest compressor either with or without the use of a surgical mask until rescuers with adequate PPE for AGPs are able to relieve the patient. Chest compressions must not be interrupted for the collection and administration of a mask and face covering for both the patient and the physician given the limited rate of recorded dissemination to health care providers so far. While data in this area are still developing, once compressions have begun and personnel with suitable PPE for AGPs arrive, providers may choose to wear masks [87]. The airway is usually blocked in the unconscious victim with minimal airflow throughout chest compressions unless deliberate attempts are made to keep it open.

Because not every resuscitation room supports “negative pressure ventilation”, locking the door may assist to prevent contamination of nearby indoor places. Taking actions to adequately ventilate a restricted location, including opening windows and doors, may minimise the local accumulation of aerosols for healthcare workers in “out-of-hospital cardiac arrest”. However, if contamination of other sites in immediate proximity is not a concern, this should be performed. Furthermore, certain healthcare companies may continue to face PPE shortages, poor vaccination rates among employees, and staffing constraints.

### 8.3. Deflection of Respiratory Particles

It is uncertain if defibrillation is an AGP in and of itself. Furthermore, early animal data suggest that chest compressions after defibrillation could produce aerosol [59]. Case-control and “retrospective cohort studies” of many other infectious pathogens dispersed via aerosolization, on the other hand, suggest that the risk of transmission throughout defibrillation is low.

Exhaled respiratory particulate can flow through some oxygen-delivery masks; thus, a surgical mask on something such as a COVID-19 patient may assist in deflecting them [88]. Mask supply, on the other hand, should not potentially prevent life-saving interventions, such as “chest compressions” as well as “defibrillation”, from taking place. An endotracheal tube, supraglottic airway, or a high-efficiency particulate air (HEPA) filter on a ventilator exhaust port can catch aerosolized contaminants when effectively ventilating with bag-mask ventilation [89]. “Endotracheal intubation” should be scheduled when enough PPE-protected workers are available to perform the procedure [90].

### 8.4. Cardiopulmonary Resuscitation (CPR) and OHCA Management

Cardiopulmonary resuscitation (CPR) in suspected or diagnosed COVID-19 patients requires a different strategy than standard CPR. CPR in COVID-19 patients puts healthcare practitioners at risk [91]. This is due to the procedures that produce aerosols. It necessitates a large number of rescuers working in close proximity, which increases the risk of a personal security violation due to a high-stress occurrence. The most difficult task is to provide COVID-19 victims with the finest potential chance of survival without risking the rescuer’s life. According to the “WHO”, over 22,000 health care workers (HCWs) were exposed to COVID-19 at work around the world, accounting for between 4% and 13% of all affected patients in different countries.

Approximately 548 “HCWs” in India were contaminated by COVID-19, accounting for 1% of all infected cases. Because this can have a substantial effect on the already overburdened healthcare system, the protection of HCWs performing CPR should be prioritised. During the COVID-19 epidemic, many international institutions, including the “American Heart Association (AHA)”, the “International Liaison on Resuscitation (ILCoR)”, and the “UK Resuscitation Council”, issued interim updates and amended resuscitation recommendations. Though most of the suggestions are in agreement, there are variances in specific areas of attention among societies. Furthermore, in resource-constrained settings, it may not be possible to implement all the guidelines. For the purpose of resuscitation of COVID-19 patients, ILCoR performed a thorough study and published its results on science, treatment strategies, and task force findings. As a result, the “European Resuscitation Council (ERC)” has produced COVID-19 guidelines that include some adjustments to current standards that should be addressed during COVID-19 patient resuscitation.

#### (i) Layperson High-Quality CPR

If cardiac arrest is predicted, telecommunicators must prioritise CPR training for bystanders to provide “hands-only CPR” for an adult patient, including rescue breathing (if ready and interested) for infants and young children. Telecommunicator-CPR can significantly enhance lay rescuer CPR service and maximize a system’s survival.

There is no information on the effects of the strategies used by telecommunicators to include COVID-19 inquiries in suspected OHCA cases [92]. By introducing questions or systematic inquries regarding COVID-19, one may delay or impede lay rescuer service for all OHCA victims. This may generate worry and panic in the rescuer. In contrast, each second that CPR is prolonged, there is a substantial reduction in the chances of survival. As a result, telecommunication programmes should take COVID-19 incidence among OHCA into account [93]. If evidence suggests a prevalence rate, telecommunicators as well as rescuers should focus on chest compressions prior to asking about COVID-19.

### 8.5. Use of PPE

In an undefined OHCA, it can be difficult to tell whether an individual has COVID-19 or not. The likelihood of contacting a COVID-19-infected person who has had an OHCA will clearly vary depending on the incidence of patients in the community. Furthermore, there is only shaky proof that the most efficient life-saving operation (defibrillation) is an “aerosol generating medical procedure (AGMP)” [94]. As a consequence, reported PPE guidelines vary significantly. For example, before attending the site, all prehospital care workers must put on airborne as well as droplet PPE, or defibrillation can be performed while donning droplet precautionary PPE while, on the other hand, airborne PPE can be used when performing chest compressions as well as ventilation procedures [95]. Finally, regardless of COVID-19 condition, paramedics should put on PPE following local and regional regulations prior to contact with patients in all OHCA cases.

### 8.6. Advanced Airway Management

All airway procedures are AGMPs with a high risk. As a result, paramedics should use caution when administering airway treatment throughout the COVID-19 pandemic [96]. As per the ability and competence of paramedics, the strategy for airway management must proceed by identifying the most suitable strategy with the lowest total risk of aerosolization. A HEPA filter should be linked to the airway device. The danger of aerosol exposure will be reduced if early advanced airway management is prioritised.

### 8.7. Mechanical Chest Compressions

“Mechanical chest compression devices” may be a better option than manual chest compressions for individuals who need “prolonged resuscitation” and “chest compressions” during ambulance transport [97]. This reduces the number of people who have to perform chest compressions. As a result, contact with aerosolized particles is reduced. When compared to high-quality human chest compressions, “mechanical chest compression devices” have similar percentages of survivability to hospital discharge. “Mechanical chest compression devices” are not commonly available, and competent use necessitates hands-on training and ongoing expertise [98]. This skill is essential for limiting hands-off time when presenting the device to an OHCA individual. The goal is to keep the chest compression percentage as low as possible. As a result, these devices need only be administered by paramedics who are experts and have practice with them [99].

### 8.8. Layperson Early Defibrillation

For OHCA patients, defibrillation by laypeople in public places is a successful approach for maximising the survival benefits of early defibrillation. The danger of COVID-19 transmission to lay rescuers is unknown in relation to the use of an “automated external defibrillator” and shock. However, there is some indirect information that can be used to estimate transmission risk. Defibrillation is uncommon among cardiac arrest treatments in that it requires very little patient contact. Rescuers can provide therapy without contacting the patient aside from applying sticky electrodes [92].

### 8.9. Recommended Guidelines and Suggestions for OHCA during COVID-19

Figure 5 shows the algorithm for OHCA during the COVID-19 pandemic. Comprehensive OHCA management and treatment guidelines would assist paramedics in determining crucial interventions for patients with significantly high chances of survival. These would also restrict interventions for people whose chances of survival are slim. These suggestions may also aid in reducing the strain on hospital limited critical care facilities. Regarding the COVID-19 pandemic, studies propose a balanced strategy for all adult OHCAs with an emphasis on paramedic protection and survival in patients. The following guidelines apply to OHCA with a suspected cardiac cause [99]:To begin, the healthcare practitioner should use a surgical mask or fabric to cover his or her own and the patient’s nose and mouth.The absence of “carotid artery pulsations (CAP)”, responsiveness, and normal breathing could all be signs of cardiac arrest.To hear or feel for breathing, healthcare staff should not put their ear or cheek close to the patient’s lips.An ambulance with an “automated external defibrillator (AED)” should be summoned at the same time. Moreover, the healthcare provider should be alerted about the patient’s COVID-19 status and associated hazards. It should also be stated whether the patient is from a hotspot or a cluster location.Then, at a pace of 100–120 per minute, one must start chest compressions in the centre of the chest. No one should use a pocket mask or mouth-to-mouth resuscitation.If an AED is available, the patient should then be defibrillated (according to the rhythm) as quickly as possible to avoid brain damage. Certain measures must be observed during the patient’s travel and arrival.The situation of COVID-19 should be communicated to all new healthcare practitioners who will be in close contact with patients.Before administering CPR on a suspected COVID-19 patient, most recommendations for out-of-hospital cardiac arrest urge that healthcare providers wear full PPE.However, in developing nations, complete PPE might not even be available outside the hospital. In such instances, donning a three-ply mask with a full seal, face shield, facemask, gloves, and a plastic apron are all recommended precautions to take before doing CPR.After donning a face mask, the local caregiver (who could have previously been exposed) should be persuaded to begin CPR.If a telecommunications system is available, caregiver CPR can indeed be performed with the assistance of an HCW who directs and simulates CPR procedures. One must keep in mind that a patient’s face is wrapped with a cloth or mask before beginning CPR. Likewise, after resuscitation, good hand hygiene should be practised by washing hands with soap or a disinfecting alcohol-based gel.Healthcare professionals should also wear PPE kits while transporting patients, whether inside or outside the hospital.A closed circuit must be used to ventilate the patient. High-efficiency particulate air (HEPA) heat moisture exchanger (HME) filters should be included in all ventilator equipment, such as bag valve masks. In ambulances, a PPE kit or at the very least an N95 mask must be supplied. Ideally, ambulances with a separate driver area and a heating, ventilation, and air conditioning (HVAC) system need to be employed. In the ambulance, COVID-19-suspected/confirmed sufferers should not be accompanied by family or friends.They should not ride in the same vehicle as a COVID-19 patient according to the AHA.Furthermore, if return of spontaneous circulation (ROSC) is not obtained in the field, avoid moving the patient to the hospital to prevent additional prehospital and hospital personnel contact with the patient. According to the authors, one family member wearing a face mask can ride in the very same ambulance with the sufferer if he or she has been in direct contact with the patient.If an aerosol-generating technique is required, open the carrying vehicle’s back doors, but do so away from pedestrians.The HVAC system, if accessible, should be turned on. In circumstances where COVID-19 patient ventilation is necessary with a driver in the very same cabin, the outside air vents in the driver section must be opened, and reverse exhaust ventilation should be powered on to generate a “negative pressure gradient” in the patient area.

## 9. Challenges to OHCA Management during COVID-19

One challenge is determining an ethically appropriate trigger for initiating and reversing these suggested process adjustments [99]. They have a direct bearing on OHCA management and care during the COVID-19 pandemic. According to the findings, the activation trigger must be linked to a “ public health (or equivalent) authority” issuing an order requiring paramedics to wear enhanced PPE beyond standard practice. When the authority recommends a return to ordinary practice, revocation ought to be the trigger. This type of predictable “on/off” trigger is adaptable on a local, regional, or country level. It has the capability of seamlessly transitioning in each way [99]. When a massive surge of COVID-19 patients risks paramedic protection or threatens to overburden critical care facilities, an alternative method is to adapt. When these two dangers have passed, the revocation will occur.

As another challenge, survival of patients and recovery after OHCA is becoming more acknowledged as a critical priority for enhancing the quality of care for sufferers, families, and providers [100]. Emotional (nervousness, wrath, self-doubt), psychological (perceived stress), and existential (What is next? What is this about?) domains are critical to discuss to enable successful survivorship. Furthermore, the OHCA experience affects not only patients and family members but also lay rescuers, including professionals participating in the survival chain. Debriefings and psychological assistance may be beneficial to EMS system effectiveness and EMS professional well-being. The epidemic has brought attention to the complexities of survival as well as the importance of including long-term rehabilitation and well-being for patients and responders.

## 10. Conclusions

The prognosis for OHCA has gotten worse in the COVID-19 era. Furthermore, cardiac arrests are more common at home, and bystanders utilise AEDs less often. Despite direct COVID-19 deaths, which may account for a portion of this finding, the indirect impact connected to lockdown and realignment of medical systems may account for a significant portion. Due to the global pandemic, layman and EMS methods should avoid making large changes to evidence-based resuscitation practices at this time. A major departure from evidence-based resuscitation techniques could result in a reduction in OHCA survival. Adjusting resuscitation processes to account for the likelihood of COVID-19 should be completed concurrently with regular resuscitation procedures.

## Figures and Tables

**Figure 1 healthcare-11-00189-f001:**
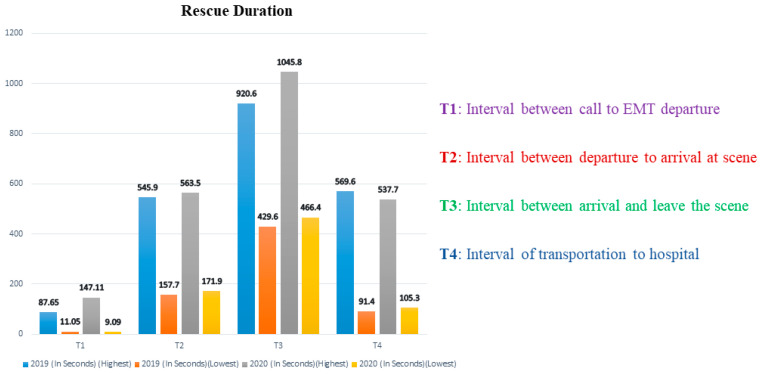
EMT rescue durations for OHCA before and during the COVID-19 pandemic [13].

**Figure 2 healthcare-11-00189-f002:**
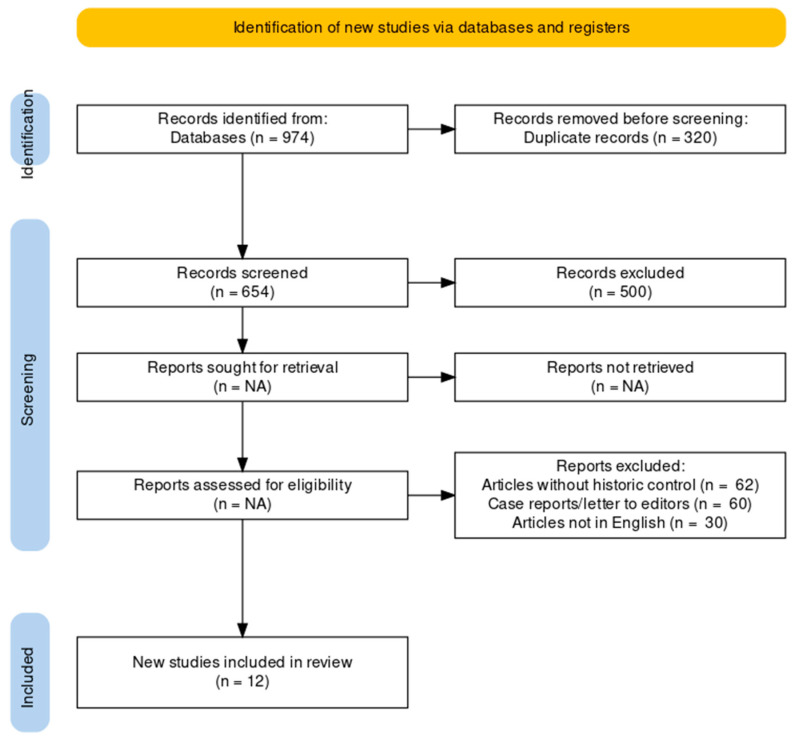
PRISMA flow chart for systematic review of OHCA during COVID-19 pandemic.

**Figure 3 healthcare-11-00189-f003:**
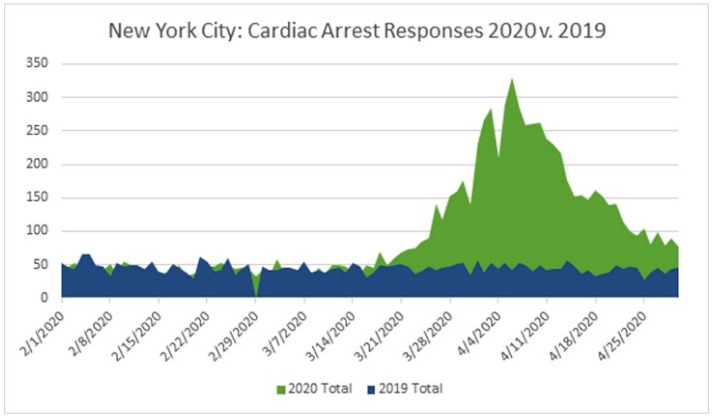
Comparison of OHCA cases before and during COVID-19 [34].

**Figure 4 healthcare-11-00189-f004:**
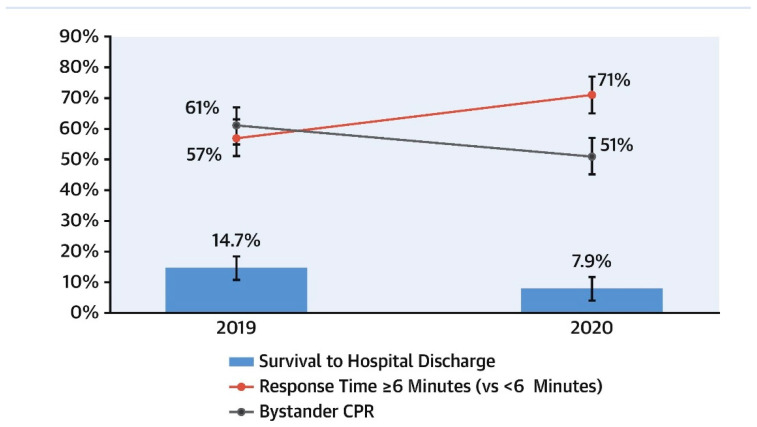
Impact of the COVID-19 for OHCA cases, on EMS response time, bystander CPR, and survival to hospital discharge [36].

**Figure 5 healthcare-11-00189-f005:**
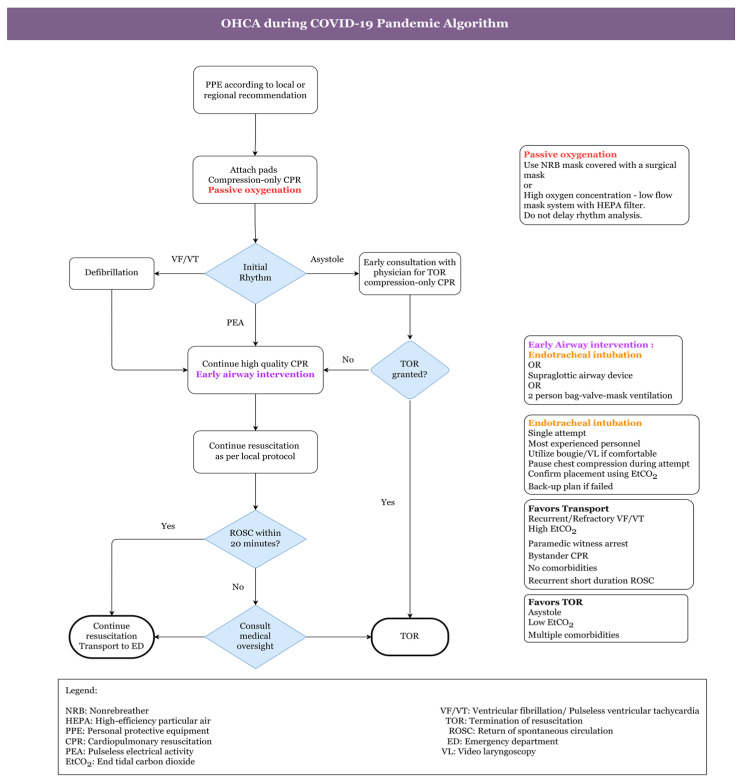
Algorithm for OHCA during COVID-19 Pandemic [99].

**Table 2 healthcare-11-00189-t002:** Site and rate of OHCA and bystander CPR in 2020 as compared with 2019 [37].

Location	2019 (*n* = 884) Number of Arrests	2019 Bystander CPR	2020 (*n* = 1034) Number of Arrests	2020 Bystander CPR
Home/residence	642 (72.6%)	301 (46.9%)	727 (70.3%)	337 (46.6%)
Street or highway	33 (3.7%	5 (15.2%)	38 (3.7%)	10 (26.3%
Place of business	46 (5.2%)	17 (37.0%)	52 (5.0%)	22 (42.3%)
Assisted living	21 (2.4%)	12 (57.1%)	29 (2.8%)	23 (79.3%)
Bus station	1 (0.1%)	1 (100.0%)	0 (0.0%)	0 (0.0%)
Dialysis	9 (1.0%	6 (66.7%)	6 (0.6%)	4 (66.7%)
Doctor’s office/clinic	8 (0.9%)	5 (62.5%)	10 (1.0%)	7 (70.0%)
Other specified place	3 (0.3%)	2 (66.7%)	2 (0.2%)	1 (50.0%)
School	4 (0.5%)	1 (25.0%)	2 (0.2%)	0 (0.0%)
Police/jail	6 (0.7%)	4 (66.7%)	9 (0.9%)	4 (44.4%)
Public building	2 (0.2%)	1 (50%)	1 (0.1%)	1 (100.0%)
Rehabilitation centre	2 (0.2%)	2 (100.0%)	10 (1.0%)	8 (80.0%)
Nursing home	97 (11.0%)	73 (75.3)	139 (13.4%)	110 (79.1%)
Religious institution	1 (0.1%)	0 (0.0%)	4 (0.4%)	3 (75.0%)
Residential institution	0 (0.0%)	0 (0.0%)	3 (0.3%)	2 (66.7%)

**Table 3 healthcare-11-00189-t003:** Some OHCA characteristics prepandemic versus COVID-19 pandemic [36].

	Prepandemic March–May 2019 (*n* = 231)	Pandemic March–May 2020 (*n* = 278)
Age, yrs	69.1 ± 17.4	64.9 ± 18.3
<35	9 (4%)	15 (5%)
35–64	76 (33%)	122 (44%)
65–84	108 (47%)	99 (36%)
≥85	38 (16%)	42 (15%)
Males	137 (60%)	174 (63%)
Witnessed arrest	122 (53%)	140 (50%)
Shockable rhythm (VF/VT)	64 (28%)	64 (23%)
Bystander CPR	142 (61%)	141 (51%)
Bystander use of AED	12 (5.2%)	4 (1.4%)

## Data Availability

The data presented in this study are available on request from the corresponding author.

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
