# Peer review of "Out-of-Hospital Cardiac Arrest during the COVID-19 Pandemic: A Systematic Review"

_healthcare, 2023, doi:10.3390/healthcare11020189_

Round 1

Reviewer 1 Report

Thank you for the opportunity to review this interesting and important manuscript.

The authors reviewed the implications of the COVID-19 pandemic on “Out of Hospital Cardiac Arrest” incidence and strategies of delivering help by both bystanders and the healthcare system.

I would suggest that the authors use proofreading services since some English-style corrections are needed.

In search methodology authors state that  The Preferred Reporting Items for Systematic Reviews and Meta-Analyses (PRISMA) standards were followed in compiling the systematic review, however scarce items from PRISMA 2020 Checklist were applied.

The authors introduced many abbreviations in the manuscript, but quite often did not define them with the first use, which is confusing to the reader; I would suggest carefully reviewing the manuscript and introducing only absolutely necessary abbreviations and defining them with the first use.

The authors stated that they used Google Scholar for the extraction of papers but it is not included in the search strategy (in Figure 2), since more than one database was used in the literature search process the authors should include information about how they handled duplicates, moreover, the sequence of some steps in the search strategy seems odd – eg full text screening before title and abstract screening step or inclusion papers form citations after checking eligibility criteria – when the authors decided to use the PRISMA flow diagram it would be rational to follow this procedure more strictly.

I would also suggest including the relevant country (or City if applicable) for each paper included in Table 1, also when describing results of particular studies it would be beneficial to mention to which country the outcomes refer.

The authors should clearly define the variables provided in Table 3 (are those numbers or percentages ect.)

Author Response

Comment

Response

Thank you for the opportunity to review this interesting and important manuscript.

The authors reviewed the implications of the COVID-19 pandemic on “Out of Hospital Cardiac Arrest” incidence and strategies of delivering help by both bystanders and the healthcare system.

I would suggest that the authors use proofreading services since some English-style corrections are needed.

We appreciate the worthy reviewer for positive comments. The manuscript has been proofread for English style corrections by expert proofreader.

In search methodology authors state that The Preferred Reporting Items for Systematic Reviews and Meta-Analyses (PRISMA) standards were followed in compiling the systematic review, however scarce items from PRISMA 2020 Checklist were applied.

The methodology has been improved with clarity in the use of PRISMA guidelines.

The authors introduced many abbreviations in the manuscript, but quite often did not define them with the first use, which is confusing to the reader; I would suggest carefully reviewing the manuscript and introducing only absolutely necessary abbreviations and defining them with the first use.

We thank to the reviewer for citing this anomaly. The whole manuscript has been checked for proper use of abbreviations as suggested.

The authors stated that they used Google Scholar for the extraction of papers but it is not included in the search strategy (in Figure 2), since more than one database was used in the literature search process the authors should include information about how they handled duplicates, moreover, the sequence of some steps in the search strategy seems odd – e.g. full text screening before title and abstract screening step or inclusion papers form citations after checking eligibility criteria – when the authors decided to use the PRISMA flow diagram it would be rational to follow this procedure more strictly.

We thank the reviewer for this suggestion. The search strategy has been updated with modified Figure 2 following PRISMA guidelines. The protocol for the removal of duplicates has also been added as suggested by the worthy reviewer.

I would also suggest including the relevant country (or City if applicable) for each paper included in Table 1, also when describing results of particular studies it would be beneficial to mention to which country the outcomes refer.

We thank the reviewer once again for this suggestion. The country for each article has been mentioned in Table 1 as suggested by the reviewer.

The authors should clearly define the variables provided in Table 3 (are those numbers or percentages etc.)

We understand the inconvenience caused to the reviewer. We have clearly defined the variables in Table 3in the revised version of the manuscript.

Reviewer 2 Report

Thank you for the opportunity to review this manuscript.

I have several suggestions to authors.

First, the title includes “A review.” What kind of review should it be? Is it narrative? systematic? 

It seems that this manuscript is somewhere between narrative and systematic.

If this is a systematic review, I wonder if the authors followed PRISMA statement.

Second, the literature search result is not fully described. Figure 2 appears first in “Search methodology.” The figure shows how many literatures were excluded. It should be described in results section. Further, authors need to describe the reasons to exclude these literatures.

My biggest concerns are on Figure 2 and 3. These figures are from the references (#52, #63).

Please forgive my ignorance regarding permission to use original figures. In my understanding, it should be presented with “adapted” or “based on” and copyright information. 

Minor points

What is CRF? I could not find what CRF stands for.

Author Response

Comment

Response

First, the title includes “A review.” What kind of review should it be? Is it narrative? systematic? 

It seems that this manuscript is somewhere between narrative and systematic.

If this is a systematic review, I wonder if the authors followed PRISMA statement.

We thank the reviewer for investing ample time to review this article. The title has been revised with the word “systematic review” as we used a systematic literature search strategy.

Second, the literature search result is not fully described. Figure 2 appears first in “Search methodology.” The figure shows how many literatures were excluded. It should be described in results section. Further, authors need to describe the reasons to exclude these literatures.

We thank the reviewer for this suggestion. Figure 2 has been modified as suggested by the reviewer.

My biggest concerns are on Figure 2 and 3. These figures are from the references (#52, #63).

Please forgive my ignorance regarding permission to use original figures. In my understanding, it should be presented with “adapted” or “based on” and copyright information.

We thank the reviewer once again for this advice. Figure 2 has been prepared from PRISMA online flow chart diagram. For Figure 3 Copyright Permission has been collected from the publisher (Elsevier).

Minor points

What is CRF? I could not find what CRF stands for.

Thanks to the reviewer for their concern, however; the abbreviation CRF is not used in the manuscript. Abbreviation CFR was defined as “Community first responder”.

Round 2

Reviewer 2 Report

Thank you for considering my suggestions. The manuscript is well revised.